# Association between Proton Pump Inhibitor Use and Parkinson’s Disease in a Korean Population

**DOI:** 10.3390/ph15030327

**Published:** 2022-03-09

**Authors:** Ji-Hee Kim, Jae-Keun Oh, Yoo-Hwan Kim, Mi-Jung Kwon, Joo-Hee Kim, Hyo-Geun Choi

**Affiliations:** 1Department of Neurosurgery, Hallym University College of Medicine, Anyang 14068, Korea; kimjihee.ns@gmail.com (J.-H.K.); ohjaekeun@gmail.com (J.-K.O.); 2Department of Neurology, Hallym University College of Medicine, Anyang 14068, Korea; drneuroneo@gmail.com; 3Department of Pathology, Hallym University College of Medicine, Anyang 14068, Korea; mulank@hanmail.net; 4Division of Pulmonary, Allergy, and Critical Care Medicine, Department of Medicine, Hallym University College of Medicine, Anyang 14068, Korea; luxjhee@gmail.com; 5Hallym Data Science Laboratory, Department of Otorhinolaryngology-Head & Neck Surgery, Hallym University College of Medicine, Anyang 14068, Korea

**Keywords:** nested case–control study, neurodegeneration, Parkinson’s disease, proton pump inhibitors

## Abstract

Few studies have shown an increased risk of Parkinson’s disease (PD) with the use of proton pump inhibitors (PPIs), and the pathophysiological mechanism for this association has not been unveiled. This study examined the relationship between PPI use and PD in a Korean population. We investigated 3026 PD patients and 12,104 controls who were matched by age, sex, income, and region of residence at a ratio of 1:4 in the Korean National Health Insurance Service, National Sample Cohort between 2002 and 2015. We estimated the associations between current and past use of PPIs and PD using odds ratios (ORs) and 95% confidence intervals (CIs) in a conditional/unconditional logistic regression after adjusting for probable confounders. Compared with PPI nonusers, both current users and past users had significantly greater odds of having PD, with ORs of 1.63 (95% CI = 1.44–1.84) and 1.12 (95% CI = 1.01–1.25), respectively. A significant association with PD was observed in individuals who used PPIs for 30–90 days and ≥90 days (OR = 1.26 and 1.64, 95% CI = 1.12–1.43 and 1.43–1.89) but not among those who used PPIs for <30 days. Both current and past use of PPIs associated with a higher probability of PD in the Korean population. Our study provides evidence regarding the association between PPI exposure and PD, but further investigation and possible explanations are warranted.

## 1. Introduction

Parkinson’s disease (PD) is an age-related progressive neurodegenerative disorder characterized mainly by dopamine-producing neuronal loss in the substantia nigra in the midbrain, and it produces a broad spectrum of motor and nonmotor manifestations [1]. Although much progress has been made in comprehension the etiopathogenesis of PD, the primary cause of neuronal death in PD remains elusive. A variety of factors, including genetic predispositions and/or environmental exposure to toxins, might be etiologies triggering the onset of PD. Among environmental factors, it is accepted that several drugs commonly prescribed, including nonsteroidal anti-inflammatory drugs (NSAIDs), calcium channel blockers (CCBs), and statins, are associated with PD risk [2].

Proton pump inhibitors (PPIs) are broadly prescribed to manage numerous gastrointestinal disorders, including gastric or duodenal ulcer disease, gastroesophageal reflux disease (GERD), *Helicobacter pylori* infection, and pathological hypersecretory conditions, and to prevent gastrointestinal bleeding in patients receiving antiplatelet therapy or taking NSAIDs [3]. Generally, PPIs are well tolerated with few adverse effects; however, their safety with long-term use has recently raised major concern [3]. A great deal of evidence from the past several years has suggested that PPI therapy is associated with a higher risk of numerous adverse effects, such as cardiovascular disease, both acute and chronic kidney disease, hypomagnesemia, *Clostridium difficile* infection, pneumonia, dementia, upper gastrointestinal cancer, and osteoporotic fractures [4].

Although PPI use has not been proposed as a contributing factor to PD pathogenesis thus far, it may be worth investigating the relation between PPI use and PD based on several pharmacoepidemiological reports concerning the effect of PPI use on increasing the risk of Alzheimer’s disease (AD), another neurodegenerative disease. When we explored the PubMed and Embase databases for studies that reported on the association between PPI use and PD using the keywords “((Parkinson’s disease) AND (proton pump inhibitor))” and limited the consequences to articles published in English and human-based studies published before October 2021, only two epidemiological studies were found that examined this association, and the results indicated a significant relation between PPI use and PD [5,6]. Moreover, no feasible mechanisms elucidating the association between exposure to PPIs and a higher risk of PD have been reported. Although neuroprotective roles of PPIs through some anti-inflammatory and antioxidant effects have been recently reported, the association between PPI use and a greater risk of developing PD is also conceivable considering that PPIs can obstruct ionic pumps in the central nervous system (CNS) and promote pathological conditions associated with minimized pH in the brain, cerebrospinal fluid, and blood [7] and that the pharmacological effects of PPIs also appear to have an effect on the development of PD.

Considering the results of the previous reports comprehensively, we hypothesized that there might be a positive correlation between PPI use and PD occurrence, but we also did not exclude the possibility of a negative correlation. The objective of the current study was to identify the association between PPI exposure and PD occurrence in a Korean population using a national health screening cohort. Furthermore, we discuss the plausible mechanism explaining why PPI use may increase the probability of PD.

## 2. Results

Most general features were balanced after matching, although there were minor differences in the baseline Charlson comorbidity index (CCI), history of head trauma or other neurodegenerative diseases of the nervous system, GERD, and the duration of H_2_ blocker use (each standardized difference (SD) > 0.2, Table 1).

The fully adjusted odds ratio (OR) in Model 3 for PD in current users of PPIs compared with that in nonusers was 1.63 (95% confidence interval (CI) = 1.44–1.84, *p* < 0.001). Compared with nonusers, past users of PPIs had a statistically significant probability of PD of 1.12 (95% CI = 1.01–1.25, *p* = 0.035). Although participants who used PPIs for <30 days did not have a statistically significant probability of PD relative to nonusers, participants who used PPIs for 30–90 and ≥90 days had a significant 1.26- and 1.64-fold probability of PD (95% CI = 1.12–1.43, *p* < 0.001 and 95% CI = 1.43–1.89, *p* < 0.001, respectively). All durations of PPI use were related to a significantly elevated likelihood of having PD in analyses for first-generation PPIs (for durations of PPI use of <30, 30–90, and ≥90 days; fully adjusted OR = 1.27, 1.41, and 1.52, 95% CI = 1.13–1.43, 1.22–1.63, and 1.27–1.80, respectively, all *p* < 0.001), but the use of PPIs for ≥90 days was significantly related to a higher likelihood of having PD in analyses for second-generation PPIs (fully adjusted OR = 1.45, 95% CI = 1.17–1.79, *p* < 0.001, Table 2).

Subgroup analyses stratified by various covariates and adjusted ORs of the associations of PPI use with PD relative to nonusers were generally consistent with the primary results (Appendix A). Additionally, the calculated ORs of PPI use for ≥30 days compared to PPI use for <30 days and those stratified by various covariates also were not significantly different from the primary results (Appendix A). Subgroup analyses stratified by covariates of the ORs for PD per 90 days of PPI use remained more likely for both first- and second-generation PPI use (Appendix A).

## 3. Discussion

In this study, we found that PPI exposure regardless of current use or past use had a greater association with PD than nonuse. In addition, our results demonstrated that these associations were significant in the case of PPI use for 30–90 days and for ≥90 days in the analyses of the duration of PPI use.

As stated above, only two studies have explored the impact of PPI therapy on the risk of PD before the present study [5,6]. One population-based case–control study of 4280 PD and control participants more than 65 years old in Taiwan detected a significant association between PD and PPI use (OR = 1.15, 95% CI = 1.04–1.27) [5]. Similarly, another population-based case–control study of 4484 patients with a first PD diagnosis in Denmark indicated that PPI therapy prior to the diagnosis of PD for 5 years or longer remained associated with PD (OR = 1.23, 95% CI = 1.11–1.37) [6]. Because the authors in these studies have not proposed hypotheses or a pathophysiological mechanism for their findings, there is no mechanism to date elucidating that PPI use could increase the probability of PD. With regard to this association, possible explanations could be divided into direct mechanisms due to pharmacological actions and indirect mechanisms through the adverse effects of PPIs.

A possible direct pharmacological effect may be defective lysosomal function and the disruption of cholinergic systems, which could infer a probable role in the pathogenesis of PD based on several recent observations. First, PPIs may affect the modulation of vacuolar-type H^+^–adenosine triphosphatase (V-ATPase), an intracellular ATP-driven proton pump that acidifies intracellular compartments. Accordingly, altered V-ATPase activity and lysosomal pH dysregulation compromise required autophagy and are linked to a broad range of proteinopathies related to cellular aging and adult-onset neurodegenerative diseases, including forms of PD and AD [8]. Second, associations between PPI use and PD may also be attributed to changes in the cholinergic system. It is known that a disruption of cholinergic interneurons, which closely interact with dopaminergic afferent neurons from the midbrain, often results in basal-ganglia-related movement disorders such as PD [9,10]. One recent report provided convincing evidence that PPIs could constrain choline-acetyltransferase, the crucial cholinergic enzyme responsible for the biosynthesis of cholinergic signaling substances [11]. However, further studies are needed to determine whether these direct mechanisms are applicable in the PD model since the findings were mainly observed in the AD model.

Another possible mechanism derived from the adverse effects of PPIs involves alterations in gut microbiota composition and magnesium and iron deficiencies, which may indirectly influence PD development. First, PPIs have been described to change the gut microbiome and elevate the number of *Streptococcus* organisms, which densely colonize the oral cavity [12]. Thus, dysbiosis within the normal gut microbiome is linked to pathophysiological changes in the GI system, enteric nervous system, and CNS, which are thought to ultimately cause the loss of dopaminergic neurons via various mechanisms. Recently, several studies have demonstrated the impact of such dysbiosis on the CNS and whether it is linked to other neurodegenerative disorders, such as AD, Huntington’s disease, amyotrophic lateral sclerosis, and multiple sclerosis [13,14,15,16,17]. Concerning PD, this was shown from the alteration in gut microbiota composition confirmed in fecal samples of PD patients compared to healthy controls [18,19]. In addition, one study quantitatively analyzing the concentrations of short-chain fatty acids, one main metabolic product of certain gut bacteria, showed that the production of short-chain fatty acids was reduced in PD patients compared with age-matched controls [19]. Second, PPI-induced magnesium deficiency can increase calcium influx and excitotoxicity, the course of which leads to neuronal dysfunction [20]. Magnesium also has neuroprotective properties through the regulation of oxidative stress by suppressing reactive oxygen species production and decreasing lipid peroxidation [21]. Although the etiological mechanism of PD related to magnesium deficiency is still poorly understood, a possible association can be speculated through the neuroprotective effect of magnesium shown in previous experimental and clinical studies [22,23]. Third, long-term PPI administration may reduce gastric acid secretion, which results in impaired iron absorption and consequently could reduce the level of iron. Given that tyrosine hydroxylase, an enzyme responsible for dopamine synthesis, is iron-dependent [24] and iron deficiency impairs dopamine reuptake in a mouse model [25], iron deficiency is a side effect of long-term PPI treatment that is possibly associated with PD development.

Our results revealed that the association between PPI and PD was significant in the case of PPI use for more than 90 days, as well as 30–90 days in the analyses of the duration of PPI use. Notwithstanding limited evidence, a clinical study and case reports have shown that not only long-term exposure to PPI but also short-term use seems to have some effect on the CNS, such as various cognitive functions, delirium, or psychotic symptoms [26,27,28,29].

Contrary to our findings and possible explanations, some prior reports have suggested that PPIs may exert neuroprotective effects via anti-inflammatory, carbonic anhydrase inhibitory, antioxidant, and apoptotic control activities [30,31]. Specifically, lansoprazole and omeprazole have been shown to have anti-inflammatory effects by attenuating human neutrophil adherence to endothelial cells in an in vitro model [32] and inhibiting the production of oxygen-derived free radicals in an in vitro model [33]. In addition, other in vitro experiments verified that omeprazole restrains the release of tumor necrosis factor-alpha and interleukin-6 from human microglial cell culture and human monocyte culture [34]. In aggregate, several studies have recently been performed on the neuroprotective effects of PPIs, but available evidence is still limited. Therefore, the more exact mechanisms for the link between PPI use and the increased probability of PD remain to be elucidated, and further investigation in preclinical and clinical studies will be necessary to conclude clear relationships.

Our results should be interpreted in the context of several limitations. First, we cannot certify drug compliance, namely, adherence to the PPI prescription, because we calculated the individual’s duration of PPI use depending on the prescription information from a claims database. Second, people who frequently visit clinics for GI disorder treatment could have a greater chance of being checked for other diseases, including PD. Furthermore, people who have taken PPIs may be more likely to have already been diagnosed with other diseases or to be taking other drugs, which might be implicated in the contribution to PD development. Specifically, these unconsidered confounding factors included the prescription of NSAIDs, which are well-known to have a protective role against PD, the co-use of antiplatelet or anticoagulant agents, and other GI disorders that trigger PPI administration. These might confound the PPI-PD relationship but are not available in the current analysis. Third, we only addressed exposure to PPIs over 1 year; thus, we could not confirm a more prolonged effect of PPIs on PD. Moreover, we cannot rule out the possibility that participants actually took PPIs for a longer period than 1 year before the index date. In practice, growing concerns about inappropriate indications and potential overuse are confirmed by several observational reports on PPI use for longer durations than recommended by clinical guidelines [35]. Finally, we cannot exclude the possibility of residual confounding in the main analyses because information on several potential confounders (other medications with or without interaction, lifestyle, and diet) was not considered in this study. Nonetheless, our study provided additional evidence regarding the association between PPI exposure and PD through a large-scale database.

## 4. Materials and Methods

### 4.1. Ethics

Ethics approval of the study protocol was obtained from the ethics committee of Hallym University (2019-10-023), and the requirement for written informed consent was waived by the Institutional Review Board.

### 4.2. Study Population

For this study, data from the Korean National Health Insurance Service, Health Screening Cohort were obtained. The information on the data was provided in detail previously [36]. We recruited PD participants from among 514,866 participants with 615,488,428 medical claim codes 2002–2015 (*n* = 6483). To select the control group, individuals who had never been diagnosed with PD among the same participants during the same period were initially extracted from the database (*n* = 508,383).

### 4.3. Proton Pump Inhibitors (Exposure)

The database used in this study includes health insurance claim codes (procedures and prescriptions) for almost all citizens. Accordingly, we can collect information on PPI prescriptions from the database using prescription codes, including the duration of use and types. Participants were classified according to their exposure to PPIs as nonusers, current users, and past users. We defined “current users” if the time of PPI use was between 0 and 30 days before the index date and “past users” if the time of PPI use was between 31 and 365 days before the index date. The duration of PPI use was calculated by summing the durations of PPI use periods for each individual for 1 year before the index date and subsequently categorized into nonuse, use for <30 days, use for 30–90 days, and use for ≥90 days. This calculation was applied for all PPIs, first-generation PPIs, and second-generation PPIs.

### 4.4. Parkinson’s Disease (Outcome)

The occurrence of PD was ascertained using the G20 code (Parkinson’s disease) of the 10th Revision of the International Classification of Diseases (ICD-10). To ensure the accuracy of diagnosis, only participants who visited clinics ≥ 2 times were regarded as having PD.

### 4.5. Participant Selection

Among PD patients, we excluded patients who were diagnosed with PD in 2002 to include only newly diagnosed cases (washout period, *n* = 364). We also excluded PD patients under 50 years old (*n* = 121) and those who had no body mass index (BMI) or total cholesterol (*n* = 5) data. Among control participants, those who had been diagnosed with PD once were excluded (*n* = 1560).

We matched the controls with four PD patients on the basis of the following basic demographics: age, sex, income, and region of residence. Control participants were designated in random number order to avoid selection bias and under the assumption that that they were studied at the same time as each matching PD patient to set the same index date. As a result of the matching procedure, 482,851 control participants were excluded from the analysis, and consequently, 5993 PD participants and 23,972 control participants were included in this study (Figure 1).

### 4.6. Covariates

Covariates obtained from records in the database included basic demographics, obesity, smoking, alcohol consumption, systolic blood pressure (SBP, mmHg), diastolic blood pressure (DBP, mmHg), fasting blood glucose (mg/dL), total cholesterol level (mg/dL), the CCI, a history of head trauma or other neurodegenerative diseases of the nervous system, GERD, and the duration of H_2_ blocker use. Age was grouped into 8 categories with 5-year intervals from the age of 50, and income was classified into 5 classes from lowest to highest. The region of residence was dichotomized into urban and rural areas following the description of a previous study [37]. Further details regarding the categorization of smoking, alcohol consumption, and obesity using BMI (kg/m^2^) were also provided in our prior work [38]. The CCI was used to evaluate 17 comorbidities and was calculated excluding dementia in this study.

As potential confounders of PD, a history of head trauma or other degenerative diseases of the nervous system was assessed using ICD-10 codes S00-S09 (diagnosed by neurologists, neurosurgeons, or emergency medicine doctors in head and neck CT (Claim codes: HA401-HA416, HA441-HA443, HA451-HA453, HA461-HA463, or HA471-HA473)) and ICD-10 codes G30-G32 (diagnosed by neurologists) only when the patient had been treated at least twice.

As potential confounders of PPI use, participants who had been diagnosed with GERD (ICD-10 code: K21, treated ≥2 times and prescribed a PPI for ≥2 weeks) and the duration of H_2_ blocker use were additionally collected based on the period of 1 year before the index date.

### 4.7. Statistical Analyses

Whether the proportions of clinical characteristics of the PD and control groups were balanced was determined using the standardized difference (SD). The associations between PPI exposure and PD were estimated as ORs and 95% CIs using conditional logistic regression models. Model 1 was adjusted for blood pressure, fasting blood glucose, total cholesterol, obesity, smoking, alcohol consumption, the CCI, and a history of head trauma or other degenerative diseases of the nervous system in the crude model, which was matched by age, sex, income, and the region of residence, and Model 2 was additionally controlled for GERD and the use of H_2_ blockers in Model 1. The associations between the duration of PPI use, which was treated as a categorical variable, and PD were determined for all PPIs as well as for first- and second-generation PPIs. PPI nonuse was used as a reference in these analyses.

For subgroup analyses, the main analyses were stratified by age (<75 and ≥75 years old), sex, income (low and high), and region of residence (rural and urban) using conditional logistic regression and by obesity (underweight, normal weight, overweight, and obese), smoking (nonsmoker and smoker), alcohol consumption (<1 and ≥1 time a week), blood pressure (normal and hypertension), fasting blood glucose (<100 and ≥100 mg/dL), total cholesterol (<200 and ≥200 mg/dL), CCI (0, 1, and ≥2), a history of head trauma or other degenerative diseases of the nervous system, GERD, and the use of H_2_ blockers using unconditional logistic regression. The criterion of the age classification was the median value of all participants. Additional ORs with 95% CIs for PPI use for ≥30 days were calculated when compared with PPI use for <30 days. Furthermore, the association between the duration of PPI use, which was treated as a continuous variable, and PD occurrence was estimated for both first- and second-generation PPI use, including stratified subgroup analyses.

Differences were considered statistically significant at a *p* value of <0.05 in two-tailed analyses. All analyses were conducted with SAS version 9.4 (SAS Institute Inc., Cary, NC, USA).

## 5. Conclusions

PPI use was associated with an increase in the incidence of PD, 1.63-fold for current use and 1.12-fold for past exposure, in a Korean population. The possibility that current or past use of PPIs and the use of PPIs for more than 30 days contribute to PD development deserves further investigation.

## Figures and Tables

**Figure 1 pharmaceuticals-15-00327-f001:**
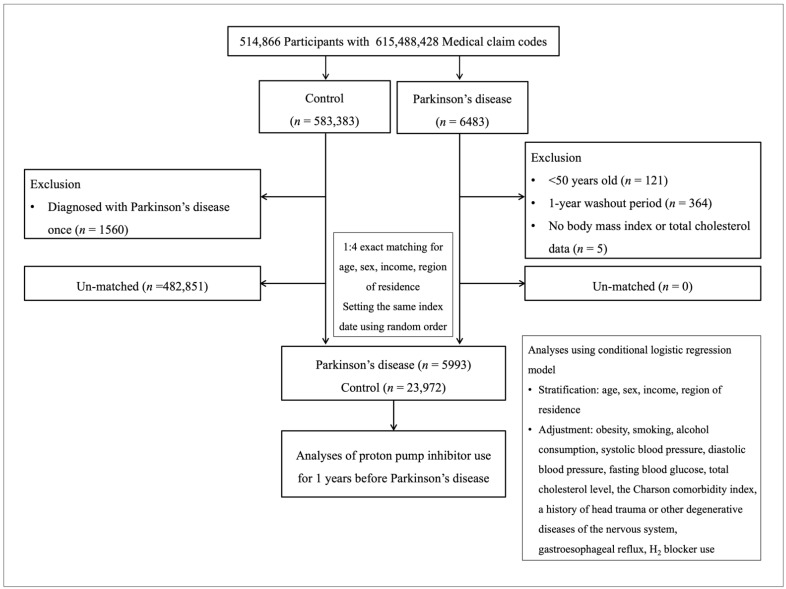
A schematic illustration of the participant selection process that was used in the present study. Of a total of 514,866 participants, 5993 of Parkinson’s disease participants were matched with 23,972 of control participants for age, sex, income, and region of residence.

**Table 1 pharmaceuticals-15-00327-t001:** General characteristics of participants.

Characteristics		Total Participants		
		Parkinson’s Disease	Control	Standardized Difference
Total number (*n*, %)		5993 (100.0)	23,972 (100.0)	
Age (years old) (*n*, %)				0.00
	50–54	234 (3.9)	936 (3.9)	
	55–59	361 (6.0)	1444 (6.0)	
	60–64	654 (10.9)	2616 (10.9)	
	65–69	1014 (16.9)	4056 (16.9)	
	70–74	1431 (23.9)	5724 (23.9)	
	75–79	1396 (23.3)	5584 (23.3)	
	80–84	721 (12.0)	2884 (12.0)	
	85+	182 (3.0)	728 (3.0)	
Sex (*n*, %)				0.00
	Male	2800 (46.7)	11,200 (46.7)	
	Female	3193 (53.3)	12,772 (53.3)	
Income (*n*, %)				0.00
	1 (lowest)	1138 (19.0)	4552 (19.0)	
	2	665 (11.1)	2660 (11.1)	
	3	814 (13.6)	3256 (13.6)	
	4	1135 (18.9)	4540 (18.9)	
	5 (highest)	2241 (37.4)	8964 (37.4)	
Region of residence (*n*, %)				0.00
	Urban	2224 (37.1)	8896 (37.1)	
	Rural	3769 (62.9)	15,076 (62.9)	
Obesity (*n*, %) ^a^				0.02
	Underweight	251 (4.2)	932 (3.9)	
	Normal	2141 (35.7)	8664 (36.1)	
	Overweight	1564 (26.1)	6255 (26.1)	
	Obese I	1847 (30.8)	7420 (31.0)	
	Obese II	190 (3.2)	701 (2.9)	
Smoking status (*n*, %)				0.09
	Nonsmokers	4733 (79.0)	18,103 (75.5)	
	Past smokers	659 (11.0)	2810 (11.7)	
	Current smokers	601 (10.0)	3059 (12.8)	
Alcohol consumption (*n*, %)				0.12
	<1 time a week	4680 (78.1)	17,436 (72.7)	
	≥1 time a week	1313 (21.9)	6536 (27.3)	
Systolic blood pressure (*n*, %)				0.02
	<120 mmHg	1351 (22.5)	5487 (22.9)	
	120–139 mmHg	2839 (47.4)	11,528 (48.1)	
	≥140 mmHg	1803 (30.1)	6957 (29.0)	
Diastolic blood pressure (*n*, %)				0.02
	<80 mmHg	2616 (43.7)	10,668 (44.5)	
	80–89 mmHg	2163 (36.1)	8609 (35.9)	
	≥90 mmHg	1214 (20.3)	4695 (19.6)	
Fasting blood glucose (*n*, %)				0.12
	<100 mg/dL	3235 (54.0)	14,165 (59.1)	
	100–125 mg/dL	1897 (31.7)	7166 (29.9)	
	≥126 mg/dL	861 (14.4)	2641 (11.0)	
Total cholesterol level (*n*, %)				0.04
	<200 mg/dL	3369 (56.2)	13,078 (54.6)	
	200–239 mg/dL	1777 (29.7)	7573 (31.6)	
	≥240 mg/dL	847 (14.1)	3321 (13.9)	
Charlson comorbidity index (*n*, %)				0.35
	0	2376 (39.6)	13,554 (56.5)	
	1	1369 (22.8)	4391 (18.3)	
	≥2	2248 (37.5)	6027 (25.1)	
A history of head trauma (*n*, %)				0.2
	Yes	477 (8.0)	817 (3.4)	
	No	5516 (92.0)	23,155 (96.6)	
A history of other degenerative diseases of the nervous system (*n*, %)				0.26
	Yes	476 (7.9)	556 (2.3)	
	No	5517 (92.1)	23,416 (97.7)	
Gastroesophageal reflux disease (*n*, %)				0.13
	Yes	1247 (20.8)	3823 (15.9)	
	No	4746 (79.2)	20,149 (84.1)	
Duration of H_2_ blocker use (mean, standard deviation)		68.67 (103.84)	39.92 (78.38)	0.31
PPI exposure (*n*, %)				0.17
	Current users	467 (7.8)	1010 (4.2)	
	Past users	562 (9.4)	1817 (7.6)	
Duration of PPI use (*n*, %)				0.18
	<30 days	578 (9.6)	2080 (8.7)	
	30 to 90 days	428 (7.1)	1228 (5.1)	
	≥90 days	409 (6.8)	867 (3.6)	
Duration of PPI use (first-generation PPIs) (*n*, %)				0.17
	<30 days	447 (7.5)	1323 (5.5)	
	30 to 90 days	312 (5.2)	777 (3.2)	
	≥90 days	233 (3.9)	499 (2.1)	
Duration of PPI use (second-generation PPIs) (*n*, %)				0.1
	<30 days	281 (4.7)	982 (4.1)	
	30 to 90 days	179 (3.0)	518 (2.2)	
	≥90 days	147 (2.5)	336 (1.4)	

Note: PPI—proton pump inhibitor. ^a^ Obesity (BMI, body mass index, kg/m^2^) was categorized as <18.5 (underweight), ≥18.5 to <23 (normal), ≥23 to <25 (overweight), ≥25 to <30 (obese I), and ≥30 (obese II).

**Table 2 pharmaceuticals-15-00327-t002:** Crude and adjusted odds ratios of proton pump inhibitor use for Parkinson’s disease.

Characteristics		*n* of PD	*n* of Controls	Odds Ratio for PD (95% Confidence Interval)					
		(Exposure/ Total, %)	(Exposure/ Total, %)	Crude ^b^	*p* -Value	Model 2 ^b,c^	*p* -Value	Model 3 ^b,c,d^	*p* -Value
PPI exposure									
	Current users	467/5993 (7.8%)	1010/23,972 (4.2%)	1.98 (1.76–2.22)	<0.001 ^a^	1.96 (1.74–2.20)	<0.001 ^a^	1.63 (1.44–1.84)	<0.001 ^a^
	Past users	562/5993 (9.4%)	1817/23,972 (7.6%)	1.32 (1.20–1.46)	<0.001 ^a^	1.31 (1.18–1.45)	<0.001 ^a^	1.12 (1.01–1.25)	0.035 ^a^
Duration of PPI use									
	<30 days	578/5993 (9.6%)	2080/23,972 (8.7%)	1.20 (1.09–1.33)	0.002 ^a^	1.22 (1.11–1.35)	<0.001 ^a^	1.10 (0.99–1.22)	0.0724
	30–90 days	428/5993 (7.1%)	1228/23,972 (5.1%)	1.51 (1.35–1.69)	<0.001 ^a^	1.47 (1.31–1.66)	<0.001 ^a^	1.26 (1.12–1.43)	<0.001 ^a^
	≥90 days	409/5993 (6.8%)	867/23,972 (3.6%)	2.05 (1.81–2.32)	<0.001 ^a^	2.01 (1.78–2.28)	<0.001 ^a^	1.64 (1.43–1.89)	<0.001 ^a^
Duration of PPI use (first-generation PPIs)									
	<30 days	447/5993 (7.5%)	1323/23,972 (5.5%)	1.45 (1.29–1.62)	<0.001 ^a^	1.43 (1.28–1.61)	<0.001 ^a^	1.27 (1.13–1.43)	<0.001 ^a^
	30–90 days	312/5993 (5.2%)	777/23,972 (3.2%)	1.72 (1.50–1.97)	<0.001 ^a^	1.64 (1.43–1.88)	<0.001 ^a^	1.41 (1.22–1.63)	<0.001 ^a^
	≥90 days	233/5993 (3.9%)	499/23,972 (2.1%)	2.00 (1.71–2.34)	<0.001 ^a^	1.89 (1.61–2.23)	<0.001 ^a^	1.52 (1.27–1.80)	<0.001 ^a^
Duration of PPI use (second-generation PPIs)									
	<30 days	281/5993 (4.7%)	982/23,972 (4.1%)	1.18 (1.03–1.35)	0.018 ^a^	1.22 (1.06–1.40)	0.005 ^a^	1.07 (0.93–1.24)	0.342
	30–90 days	179/5993 (3.0%)	518/23,972 (2.2%)	1.42 (1.20–1.69)	<0.001 ^a^	1.45 (1.21–1.73)	<0.001 ^a^	1.14 (0.95–1.37)	0.166
	≥90 days	147/5993 (2.5%)	336/23,972 (1.4%)	1.80 (1.48–2.20)	<0.001 ^a^	1.83 (1.49–2.23)	<0.001 ^a^	1.45 (1.17–1.79)	<0.001 ^a^

Note: PD—Parkinson’s disease; PPI—proton pump inhibitor. ^a^ Conditional logistic regression analysis, significance at *p* < 0.05; ^b^ Matched model based on age, sex, income, and region of residence; ^c^ Adjusted for systolic blood pressure, diastolic blood pressure, fasting blood glucose, total cholesterol level, obesity, smoking, alcohol consumption, the Charlson comorbidity index, and a history of head trauma or other degenerative diseases of the nervous system; ^d^ Adjusted for gastroesophageal reflux disease and H_2_ blocker.

## Data Availability

Data is contained within the article and Appendix A.

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
