# Peer review of "Association between Proton Pump Inhibitor Use and Parkinson’s Disease in a Korean Population"

_pharmaceuticals, 2022, doi:10.3390/ph15030327_

Round 1

Reviewer 1 Report

Despite several limitations of this study, which are mentioned by the authors themselves, I believe that this is an important work from the perspective of the worldwide use of PPI drugs. In addition, the authors highlight possible mechanisms that may underlie the investigated association of PPI drug use with the occurrence of parkinson's disease. 

Author Response

Despite several limitations of this study, which are mentioned by the authors themselves, I believe that this is an important work from the perspective of the worldwide use of PPI drugs. In addition, the authors highlight possible mechanisms that may underlie the investigated association of PPI drug use with the occurrence of Parkinson's disease.

We sincerely thank the reviewer for the constructive and informative comments. As the reviewer has indicated, our study has some potential limitations. However, to date, there are only two observational studies focusing on an association between PPIs and Parkinson’s disease, and the authors have not explained a plausible mechanism for this association. In contrast, we not only investigated Korean population data on the correlation between PPI use and Parkinson’s disease but also discussed the potential mechanisms to support our findings.

Reviewer 2 Report

TITLE

OK.

ABSTRACT

OK.

INTRODUCTION

OK.

METHODS

The authors state that “We collected information on PPI prescriptions”. However, more detailed information on how this information was obtained and included in the registry should be provided, as this is the most important studied factor. Thus, it should be clear how reliable the information on PPI use is in the present study.

In addition, it should be detailed how the controls were selected.

Was the dose of PPI taken into account in the analyses?

Covariates obtained from records in the database included basic demographics, obesity, smoking, alcohol consumption, systolic blood pressure, diastolic blood pressure, fasting blood glucose, total cholesterol level, the CCI, a history of head trauma or other neurodegenerative diseases of the nervous system, GERD, and the duration of H2 blocker use. However, there may be other potential risk factor for PD, that could act as confounders, and that were not considered in the present study. For example, the reason for taking PPI drugs (in addition to GERD), mainly including comorbidities or the intake of NSADs or anticoagulants, was not included. As the authors recognize, people who have taken PPIs may be more likely to have already been diagnosed with other diseases or to be taking other drugs, which might be implicated in the contribution to PD development. In this respect, the authors should clearly recognize in de Discussion section that they did not control by these potential confounders (comorbidities that are, precisely, the cause of PPI prescription).

RESULTS

Please consider deleting this sentence “section may be divided by subheadings. It should provide a concise and precise description of the experimental results, their interpretation, as well as the experimental conclusions that can be drawn”, which seems to be out of place.

DISCUSSION

Please see previous comments included in the Methods section.

Compared with PPI nonusers, past users had significantly greater odds of having PD, with OR of 1.12. However, this indicates a very small association, with a borderline 95% CI (1.01-1.25). Please comment on this in the Discussion section.

The authors found that a significant association with PD was observed in individuals who used PPIs for 30-90 days. However, it is difficult to understand how this very short time exposition (just a few weeks) could cause PD. Please comment on this in the Discussion section.

REFERENCES

OK.

TABLES

OK.

Author Response

METHODS

The authors state that “We collected information on PPI prescriptions”. However, more detailed information on how this information was obtained and included in the registry should be provided, as this is the most important studied factor. Thus, it should be clear how reliable the information on PPI use is in the present study.

We sincerely appreciate the reviewer’s constructive advice. According to the reviewer’s suggestion, we have modified the following text to more precisely express information on PPI prescription in the MATERIALS AND METHODS section in the revised manuscript as follows:

MATERIALS AND METHODS

Previous version: We collected information on PPI prescriptions, including the duration of use and types.

Revised version: The database used in this study includes health insurance claim codes (procedures and prescriptions) for almost all citizens. Accordingly, we can collect information on PPI prescriptions from the database using prescription codes, including the duration of use and types.

In addition, it should be detailed how the controls were selected.

We sincerely thank the reviewer for this observation. We have corrected the following text to help better understand the descriptions of the control group in the MATERIALS AND METHODS section in the revised manuscript:

MATERIALS AND METHODS

Previous version: We recruited PD participants from among 514,866 participants with 615,488,428 medical claim codes 2002-2015 (n = 6,483). The control group was recruited from among participants who were not defined as having PD during the same period (n = 508,383).

Revised version: We recruited PD participants from among 514,866 participants with 615,488,428 medical claim codes 2002-2015 (n = 6,483). To select the control group, individuals who had never been diagnosed with PD among the same participants during the same period were initially extracted from the database (n = 508,383).

Was the dose of PPI taken into account in the analyses?

We sincerely thank the reviewer for this valuable comment. Recently, high-dose PPI has often been prescribed, but we could not consider the dosage of PPI in this study. Although we were unable to evaluate the dosage of PPI in this study, we believe that the sum of the prescription dates, as the duration of PPI use, could be an alternative surrogate marker of PPI use.

Covariates obtained from records in the database included basic demographics, obesity, smoking, alcohol consumption, systolic blood pressure, diastolic blood pressure, fasting blood glucose, total cholesterol level, the CCI, a history of head trauma or other neurodegenerative diseases of the nervous system, GERD, and the duration of H2 blocker use. However, there may be other potential risk factor for PD, that could act as confounders, and that were not considered in the present study. For example, the reason for taking PPI drugs (in addition to GERD), mainly including comorbidities or the intake of NSADs or anticoagulants, was not included. As the authors recognize, people who have taken PPIs may be more likely to have already been diagnosed with other diseases or to be taking other drugs, which might be implicated in the contribution to PD development. In this respect, the authors should clearly recognize in de Discussion section that they did not control by these potential confounders (comorbidities that are, precisely, the cause of PPI prescription).

We sincerely thank the reviewer for the constructive and informative comments. Additionally, we completely agree with the reviewer’s comments. Thus, we have modified the following paragraph concerning the additional limitation in the DISCUSSION section of the revised manuscript:

DISCUSSION

Previous version: Furthermore, people who have taken PPIs may be more likely to have already been diagnosed with other diseases or to be taking other drugs, which might be implicated in the contribution to PD development.

Revised version: Furthermore, people who have taken PPIs may be more likely to have already been diagnosed with other diseases or to be taking other drugs, which might be implicated in the contribution to PD development. Specifically, these unconsidered confounding factors included the prescriptions of NSAIDs, which are well-known to have a protective role against PD, the co-use of antiplatelet or anticoagulant agents, and other GI disorders that trigger PPI administration. These might confound the PPI-PD relationship but are not available in the current analysis.

RESULTS

Please consider deleting this sentence “section may be divided by subheadings. It should provide a concise and precise description of the experimental results, their interpretation, as well as the experimental conclusions that can be drawn”, which seems to be out of place.

We sincerely appreciate the reviewer’s constructive advice and would like to apologize for the insertion of these inappropriate sentences. We have deleted the paragraph in the RESULTS section in the revised manuscript.

DISCUSSION

Please see previous comments included in the Methods section.

Compared with PPI nonusers, past users had significantly greater odds of having PD, with OR of 1.12. However, this indicates a very small association, with a borderline 95% CI (1.01-1.25). Please comment on this in the Discussion section.

The authors found that a significant association with PD was observed in individuals who used PPIs for 30-90 days. However, it is difficult to understand how this very short time exposition (just a few weeks) could cause PD. Please comment on this in the Discussion section.

We sincerely thank the reviewer for these valuable comments. As the reviewer has pointed out, the OR of 1.12 in past users of PPI could be regarded as indicating that this association did not have a great impact on the development of PD. In addition, the 30-90-day PPI usage period can also be considered insufficient to affect the development of PD. Although we assessed the duration of PPI use as being calculated by summing the duration of PPI use periods for each individual only for 1 year before the index date in our study, it is possible that numerous participants have taken PPI not only 1 year before the index date but also before that. In fact, growing concerns about inappropriate indications and potential overuse are confirmed by reports on increased long-term use, especially in elderly populations. In this context, one nationwide study in Iceland demonstrated a considerable increase in the overall outpatient use of PPIs over a 13-year period, and patients were increasingly treated for longer durations than recommended by clinical guidelines [Halfdanarson et al., 2018]. Therefore, we have modified the limitation regarding the effect size and the duration assessed in this study in the DISCUSSION section of the revised manuscript as follows:

DISCUSSION

Previous version: Third, we addressed exposure to PPIs over 1 year; thus, we could not confirm a more prolonged effect of PPIs on PD.

Revised version: Third, we only addressed exposure to PPIs over 1 year; thus, we could not confirm a more prolonged effect of PPIs on PD. Moreover, we cannot rule out the possibility that participants actually took PPIs for a longer period than 1 year before the index date. In practice, growing concerns about inappropriate indications and potential overuse are confirmed by several observational reports on PPI use for longer durations than recommended by clinical guidelines [35].

REFERENCES

  1. Oskar, O.H.; Anton, P.; Einar, S.B.; Sigrun, H.L.; Margret, H.O.; Eirikur, S.; Helga, M.O.; Helga, Z. Proton-pump inhibitors among adults: a nationwide drug-utilization study. Therap Adv Gastroenterol 2018, 11, 1756284818777943.

Round 2

Reviewer 2 Report

No aditional comments